# Geographic range of plants drives long-term climate change

Khushboo Gurung [1] ✉, Katie J. Field [2], Sarah A. Batterman [3,4,5], Simon W. Poulton [1] & Benjamin J. W. Mills [1]

Long computation times in vegetation and climate models hamper our ability to evaluate the potentially powerful role of plants on weathering and carbon sequestration over the Phanerozoic Eon. Simulated vegetation over deep time is often homogenous, and disregards the spatial distribution of plants and the impact of local climatic variables on plant function. Here we couple a fast vegetation model (FLORA) to a spatially-resolved long-term climate-biogeochemical model (SCION), to assess links between plant geographical range, the long-term carbon cycle and climate. Model results show lower rates of carbon fixation and up to double the previously predicted atmospheric $CO_2$ concentration due to a limited plant geographical range over the arid Pangea supercontinent. The Mesozoic dispersion of the continents increases modelled plant geographical range from 65% to > 90%, amplifying global $CO_2$ removal, consistent with geological data. We demonstrate that plant geographical range likely exerted a major, under-explored control on long-term climate change.

Atmospheric $CO_2$ is a key driver of Earth's long-term temperature[1,2]. Over the Phanerozoic Eon, atmospheric $CO_2$ concentrations have fluctuated due to a combination of tectonic and biotic events, while stabilising processes such as temperature-dependent silicate weathering have prevented runaway climates[3]. The extent to which individual biological and tectonic processes contribute towards the overall climate state is debated, and a multitude of processes working simultaneously have been proposed. However, Earth system models incorporating these processes are currently unable to adequately reproduce Phanerozoic climate change from these basic principles[1].

Vegetation structure and processes couple the atmospheric boundary layer and the land surface, as plants influence the exchange of carbon, energy and water[4,5]. Plants have a significant impact on the carbon cycle by acting both as a sink (via photosynthetic carbon fixation and silicate weathering enhancement) and source (via respiration and pyrogenic carbon from biomass burning)[6]. These vegetation-climate interactions vary in concert with local climatic conditions (e.g., temperature, water and light availability), and a change in plant

community structure and function can influence climate by controlling the rate of carbon exchange[7]. Well known 'box' models for Phanerozoic-scale biogeochemistry, such as GEOCARB[1,2,8,9] and COPSE[10,11], attempt to represent feedbacks between the plant biosphere and climate. The inclusion of land plants in these types of model has a long history[9], and the initial colonisation of the land is commonly thought to have drawn down $CO_2$ and cooled the climate in the middle to late Paleozoic Era[2,11], although this remains uncertain[12,13].

Within current long-term models, plants are restricted to influencing silicate weathering and organic matter burial via simple global relationships, without consideration of spatial heterogeneity in vegetation driven by local temperature, moisture and light regimes, even in cases where the underlying model contains a spatially-resolved land surface[14]. The use of a singular global value to represent highly heterogeneous worldwide plant interactions with local climate and biogeochemistry thus reduces the reliability of carbon cycle predictions from current models. In order to reasonably represent vegetation and its influence on biogeochemical cycles and climate over geological

[1]School of Earth and Environment, University of Leeds, Leeds, UK. [2]Plants, Photosynthesis and Soil, School of Biosciences, University of Sheffield, Sheffield, UK. [3]Cary Institute of Ecosystem Studies, Millbrook, NY, USA. [4]School of Geography, University of Leeds, Leeds, UK. [5]Smithsonian Tropical Research Institute, Panama City, Panama, USA. ✉e-mail: k.gurung@leeds.ac.uk

time, accurate biologically-mediated feedbacks need to be incorporated into long-term Earth System Models at the local scale.

Here, we combine a recently-developed spatial deep-time vegetation model 'FLORA' (Fast Land Occupancy and Reaction Algorithm[15]) with the 'SCION' (Spatial Continuous IntegratiON[14]) climate-biogeochemical model. Building on the detailed SCION representation of the impact of abiotic parameters such as runoff on global weathering, we have produced a coupled model for long-term climate change that adds both spatially-resolved and dynamic terrestrial vegetation, as well as the ability for this vegetation to impact the carbon cycle through carbon burial and weathering. This allows us to investigate how plant-climate feedbacks have affected climate regulation over geological timescales.

The SCION model uses an emulator constructed from pre-run, steady-state FOAM general circulation model (GCM) climate simulations, and calculates continental weathering rates over time on a 2D grid (40 × 48), informed by local temperature and local rates of erosion and runoff[14,16,17]. Although the treatment of weathering is more complex than in the GEOCARB or COPSE models, vegetation has remained similar to these predecessors and is represented by a single value for total global biomass, which enhances continental weathering and land-derived organic carbon burial by a given factor across the globe. In this representation, global biomass is sensitive to global average temperature and the levels of $CO_2$ and $O_2$ in the atmosphere[10].

Here, we modify the SCION model by removing the dimensionless representation of biomass and replacing it with a dynamic gridpoint-by-gridpoint linkage to the FLORA model[15], which has been streamlined and adapted from the LPJ-DGVM[18] (Lund-Postdam-Jena Dynamic Global Vegetation Model) to produce a generalised photosynthetic plant biosphere with minimal computational expense. The parameters used to estimate local rates of photosynthesis are the latitudinal distribution of solar radiation, global $O_2$ and $CO_2$ levels, and local grid cell-scale temperature and runoff. These inputs drive net primary productivity (NPP), respiration, and biomass loss. The FLORA model also includes three broad functional types of tropical, temperate and boreal vegetation, and has been shown to produce a reasonable fit to the present day distribution of plant biomass when forced with present day climatic conditions[15].

To couple the models, each continental SCION grid cell is seeded with biomass during each SCION timestep (which are approx. 100 kyrs but are continuously varied based on model stability). FLORA then runs in an internal loop until steady state is reached, such that biomass and productivity are based on the local climatic biogeochemical variables within that SCION timestep ($k$; Fig. 1). The FLORA variables are used to calculate the continental fluxes in SCION for that timestep, which define the climatic and biogeochemical conditions for SCION timestep $k + 1$, and which are then used to spin up the next iteration of FLORA. FLORA internal loops are ended once the difference between the current and the next (yearly) step biomass calculation is <1%. The calculated NPP and total vegetation at each of these approximate steady states are used to calculate the overall rate of total marine and terrestrial organic carbon ($C_{org}$) burial, and the biotic weathering enhancement at each continental grid cell (see Methods). A feedback system between vegetation and climate is thereby created, as the weathering enhancement and $C_{org}$ burial influences atmospheric $CO_2$ in SCION, and subsequently, the environmental conditions in the SCION model impact the spatial distribution of vegetation in the next global timestep ($k + 1$, Fig. 1).

## Results and discussion
The inclusion of FLORA within SCION can be used to explore the regional and global effects of vegetation on long-term biogeochemical cycles. Figure 2 highlights the distribution of modelled biomass across the model continents over selected time points through the Triassic to present day. We limit our analysis to this timeframe as FLORA does not

yet include the major plant evolutionary changes that occurred over the Paleozoic, such as the development of roots, vasculature, wood and seeds, which would likely significantly limit the biomass and geographic range of plants (e.g.,[12,19]).

## Impact of global biomass
The terrestrial biomass distribution of SCION-FLORA closely follows the distribution of continental runoff due to the strict requirements for water availability, as well as showing greater biomass closer to the equator due to higher temperatures and insolation (Fig. 2). FLORA uses runoff to infer water stress, rather than precipitation, because the FOAM climate emulator does not include precipitation data. Future models of this type should aim to use a broader spectrum of hydrological data, however, the current approach in FLORA has been shown to reproduce a reasonable approximation of the present-day distribution of biomass when forced with only runoff and temperature datasets[15]. Present-day biomass is more poorly predicted when FLORA is run within SCION, largely due to the differences between the coarse-scale GCM-modelled climate and the high resolution climatological data[20] (see Supplementary Information). For example, the FOAM emulator at 0 Ma presents a scenario more like the Last Glacial Maximum (~20 kyrs) than the present interglacial period, leading to much lower northern hemisphere temperatures than the Holocene. Nevertheless, the key tropical forested river basins (Amazon, Congo, Ganges-Brahmaputra, Yangtze, and Mississippi), as well as the latitudinal and longitudinal trends in biomass, are reasonably well represented, along with the heterogenic biotic effects on continental weathering. Given uncertainty around glacial effects on plants, our key timeframe of interest is the warm climate states of the Mesozoic, where large amplitude glacial-interglacial cycles did not occur.

The pattern of continental silicate weathering closely follows that of biomass due to the assumed biotic enhancement of silicate weathering (i.e., feedback strength; Eq. 1 in Methods), but also due to the similarity in criteria for high plant productivity and high abiotic silicate weathering (i.e., abundant runoff and high local temperatures). Similar to stand-alone runs of FLORA for paleoclimates[15], the Mesozoic Pangaea supercontinent (shown at 220 and 200 million years ago (Ma); Fig. 2) is predicted to have had large arid interior belts due to a lack of moisture transport from the oceans[16]. These arid areas do not host any plant biomass within FLORA and do not undergo silicate weathering.

The inclusion of a dynamic spatial biosphere reduces the stability of the SCION model– particularly as runaway climate-extinction events are now possible (in which the vegetation dies and the model crashes) –and model sensitivity analysis resulted in 852 successful runs across a 1000 model ensemble (see Methods). Comparison between the original SCION (without spatial vegetation) and the now SCION-FLORA (with spatial vegetation) model outputs show an overall improvement of model prediction fit to proxy data for atmospheric $CO_2$ concentration and global average surface temperature (Fig. 3–see Figure S3 for full model results for all reservoirs and fluxes). Proxy $CO_2$ concentrations generally have a large uncertainty range, and thus for clarity are plotted as central estimates without uncertainty windows. $CO_2$ levels derived from paleosols are difficult to constrain due to their sensitivity to soil-respired $CO_2$[21]. Despite this uncertainty, paleosol proxies are commonly used as they do not lose sensitivity at high $CO_2$ and can be applied throughout the Phanerozoic, unlike more recent proxies such as boron isotopes. Stomatal ratios are derived from fossil records of the *Ginkgo* family[22], and therefore rely on the quantity of fossil samples found. Due to this uncertainty, our key comparison is to the paleotemperature record, which is based on the distribution of climate-sensitive lithologies combined with oxygen isotope records[23].

## Impact of spatial vegetation
Spatially-dynamic vegetation increases model $CO_2$ concentrations and surface temperatures during the Triassic and Jurassic, while these

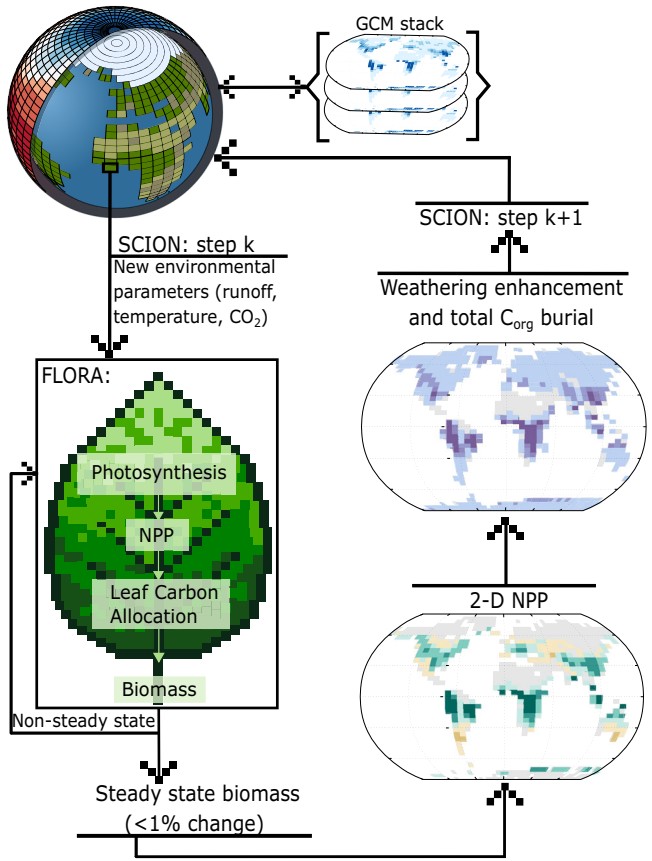

**Fig. 1 | Schematics of SCION-FLORA coupling.** The vegetation model FLORA uses SCION parameters to calculate the global distribution of biomass for each SCION timestep (k). Once FLORA reaches steady-state biomass (<1% change between the current and next biomass calculation), NPP is used to calculate weathering enhancement rates in SCION, and total vegetation biomass is used to calculate organic carbon ($C_{org}$ burial) in SCION. NPP net primary productivity, GCM General Circulation Model.

variables decrease in the Cretaceous, relative to the standard run consisting of a box-model representation of vegetation. Both of these changes bring the combined model closer in line with geological evidence for changes in surface temperatures[23] and past $CO_2$ levels (although the uncertainty in the latter is large)[22]. The cumulative error over 250 million years between proxy-derived and modelled-derived temperature is reduced by ~21% when comparing the SCION-FLORA model to the original SCION model. Habitable area (continental grid cell area that is able to support plant biomass) increases from 65% to a peak of 92% from the Triassic to Cretaceous, as a result of the break-up of Pangaea (Fig. 3D). The original SCION model (without FLORA) does not involve land area within its calculation as it treats vegetation as a global unit which is not spatially explicit. There is little difference in the Cenozoic results, which may be expected as there is little change in the model paleogeography compared to those that occurred during the Mesozoic.

Interestingly, changes to plant geographical range do not clearly track changes to total global biomass. Jurassic biomass is predicted to be higher in SCION-FLORA than in SCION alone. In the absence of other changes in previous models (e.g., COPSE), greater biomass would be expected to decrease $CO_2$ levels and thus global temperatures, yet the opposite occurs. A large geographic range of plants, and the subsequent effect on silicate weathering enhancement (Eq. 1 in Methods), produces a higher global weathering rate than a larger biomass restricted to a smaller area. Limited water circulation during the time of Pangaea created large arid areas in the subtropics—particularly

along the western continental areas that are typically dominated by descending air and easterly trade winds—and restricted a large portion of biomass to the equator[16,24] (200 Ma run, Fig. 2). As plant productivity became more isolated, less total area on the continents was subjected to biotic weathering enhancement, resulting in a weaker silicate weathering process and less $CO_2$ sequestration at a given global temperature. A clear difference in the impact of Pangea on total land area habitable by plants can be observed in Fig. 3D, as plant habitable area rises by around 20% between the Triassic and Cretaceous.

Continental fragmentation brought about more humid conditions and reorganised tropical circulation, which may have helped drive the expansion and diversification of angiosperms during the late Cretaceous[25]. In our model, relative total habitable area over the Jurassic-Cretaceous increased from ~79% to 88% of the land as Pangaea broke apart, increasing water accessibility on the continental surface[24,26]. As a result, the modelled plant biosphere expanded, allowing a wider amplification of silicate weathering across the land surface despite a drop in relative plant productivity (Fig. 3), which lowered global temperature by ~2–3 °C ($CO_2$ by ~1000 p.p.m) compared to the default model run. Migration of land away from the equatorial zone towards the northern hemisphere, as well as continental flooding[27], limits plant growth, lowering overall productivity. Thus, while the spatial representation of biomass in SCION-FLORA helps explain why SCION predicted an overly warm climate during the Cretaceous[14], it still allows for some warming driven by the movement of land away from the tropics. An increase in atmospheric $CO_2$ is observed in the model between 30 and 15 Ma, coincident with a slight decline in habitable area from $1.3 \times 10^{14}$ m² to $1.23 \times 10^{14}$ m² (Fig. 3). Movement of land towards the higher latitudes, and the loss of weathering enhancement due to aridification[28,29] in parts of Australia, South East Asia and Africa, leads to a decrease in total land $C_{org}$ burial (Figure S3C) and therefore higher $CO_2$ and warmer temperatures despite the increase in biomass around the tropics (Figure S4).

## Model limitations

Two key limitations in our approach are the lack of consideration of evolutionary events such as the angiosperm radiation, which might have altered local weathering efficiency[30], and the lack of dynamical feedback between our GCM climate emulator and the local plant biosphere—wherein plants should enhance the local hydrological cycle through transpiration[31]. Future work should aim to incorporate these mechanisms, perhaps through eco-evolutionary modelling and a more complete climate emulator which can alter grid cell-scale hydrology in response to plant habitation. However, these limitations are not likely to affect our present conclusions. Consideration of angiosperm-specific feedbacks could potentially drive a warmer pre-Cretaceous (i.e., before their evolution), which would improve the model fit to data, and although the climate emulator does not allow for changes in plant transpiration, the FOAM climate model was run with a dynamic physical vegetation model and thus the emulator gives a 'maximal' spread of water cycling considering likely transpiration effects.

It should also be noted that the model only estimates the potential long-term outcomes of vegetation on climate, and does not consider short-term biosphere processes like colonisation and biome turnover[18]. We also omit large external perturbations to the carbon cycle, such as the formation of the Central Atlantic Magmatic Province and the end-Triassic mass extinction[32], which likely lead to large turnovers of biomass. While we consider these beyond the scope of the current work, this type of coupled model should be well placed to investigate these events in the future.

Overall, environmental conditions over the Mesozoic Era—including high $CO_2$ and surface temperature—appear to have been more permissive for plant productivity than at present, allowing a higher total global land plant productivity. However, the global effect of plants on weathering rates was limited by aridity over the

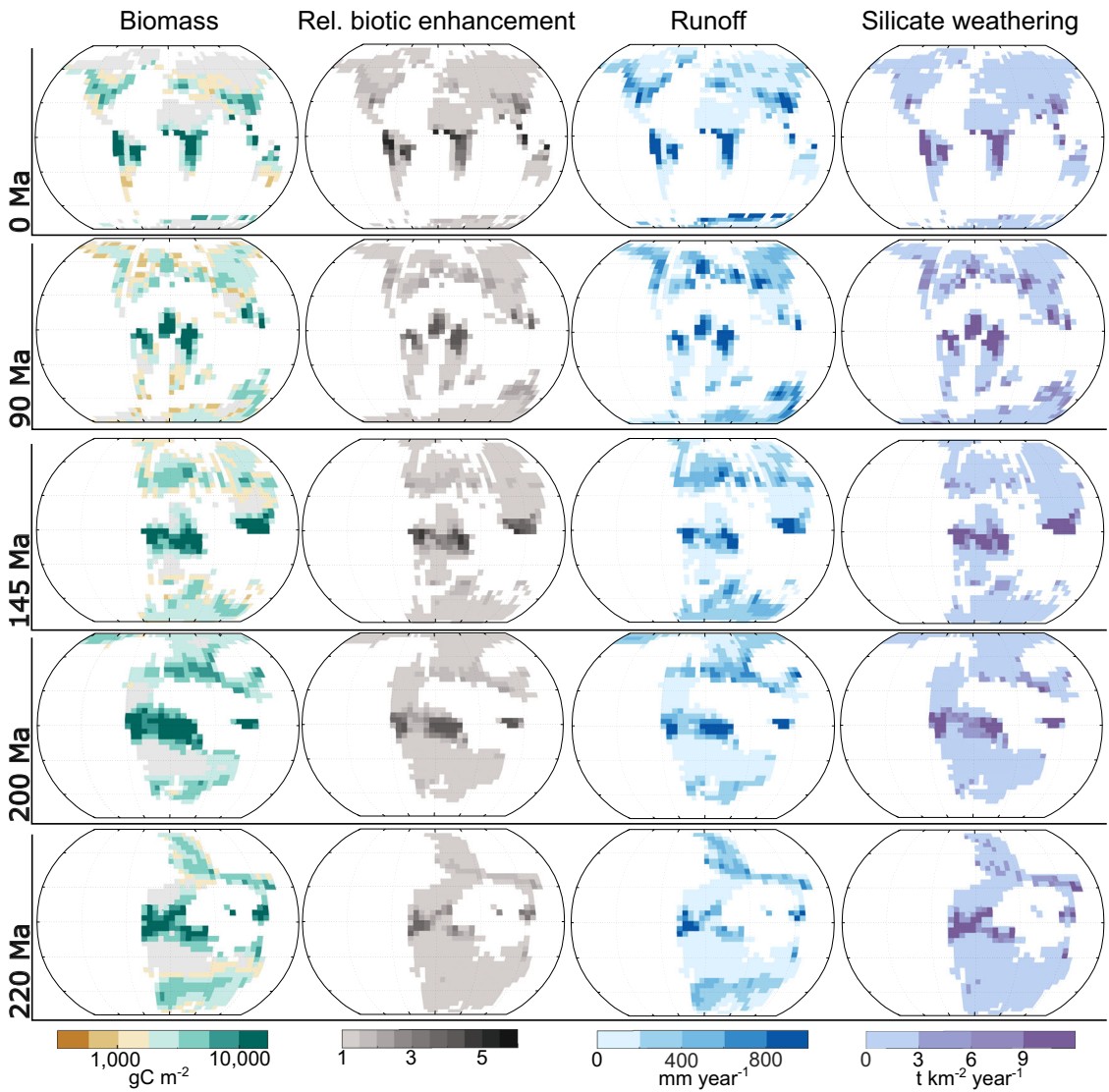

**Fig. 2 | Maps of global biomass, runoff and silicate weathering calculated by the SCION-FLORA model.** Selection of output fields to demonstrate changing habitability of the land surface and the expansion and contraction of silicate weathering zones between the Triassic and the present day. Grey areas indicate no biomass present due to no runoff or annual average temperatures being < −10 °C. The relative biotic enhancement is the factor by which silicate weathering is amplified due to the presence of plants in that grid cell, which is related to NPP in the model. Ma million years ago, NPP net primary productivity.

subtropical areas of Pangaea, which restricted plant geographic range. Using a new spatially-resolved and continuous vegetation-climate-biogeochemical model, we are able to explore aspects of the bi-directional feedbacks between plants and climate over geological time, and show the importance of the inclusion of the geographical spread of plants for weathering enhancement and hence long-term climate change. The aridity of Pangaea diminished habitable land space for plants, thereby isolating biomass within tropical regions, whereas the breakup of Pangaea had the opposite effect and led to a global spread of vegetation. Thus, even though the Mesozoic was likely a time of globally high plant biomass, the biogeochemical and climatic effects were isolated, and limited by the area of plant presence. In addition, the break-up of Pangaea created more habitable areas, ultimately suppressing the warming of Cretaceous climate, which was itself driven largely by exceptionally high volcanic carbon fluxes[27]. Our work suggests that future efforts to assess past continental weathering rates should consider the habitable land area for plants, and this may become especially important during the expansion of plants in the Paleozoic, where the biogeography of primitive and non-analogue early plants is currently particularly uncertain.

## Methods

FLORA[15] is coupled with SCION[14] (version 1.1.4) to calculate biomass at each timestep. Parameters within FLORA taken from SCION are time, temperature, runoff, continental configuration, and global $CO_2$ and $O_2$ concentration. The following details the changes made within the two models. An in-depth explanation of the FLORA model can be found in ref. 15. For every SCION timestep, FLORA runs until a near-steady state for biomass (<1% change between the subsequent biomass calculations) is reached. This creates a 2-D continental map of modelled NPP and biomass which feeds back to the silicate and carbonate weathering rates in each grid cell, along with the global rate of organic carbon burial (Fig. 1). At the beginning of each FLORA run, the terrestrial biosphere is 'seeded' at a homogenous value of $2.5 \times 10^4$ gC m$^{-2}$, which represents the average present day biomass, and accounts for re-establishment as vegetation is reset after each timestep.

### Sensitivity analysis

The sensitivity analysis of SCION-FLORA consisted of 1000 model runs, 852 of which were successful. Unsuccessful runs were largely due to initial parameter conditions leading to an unstable runaway

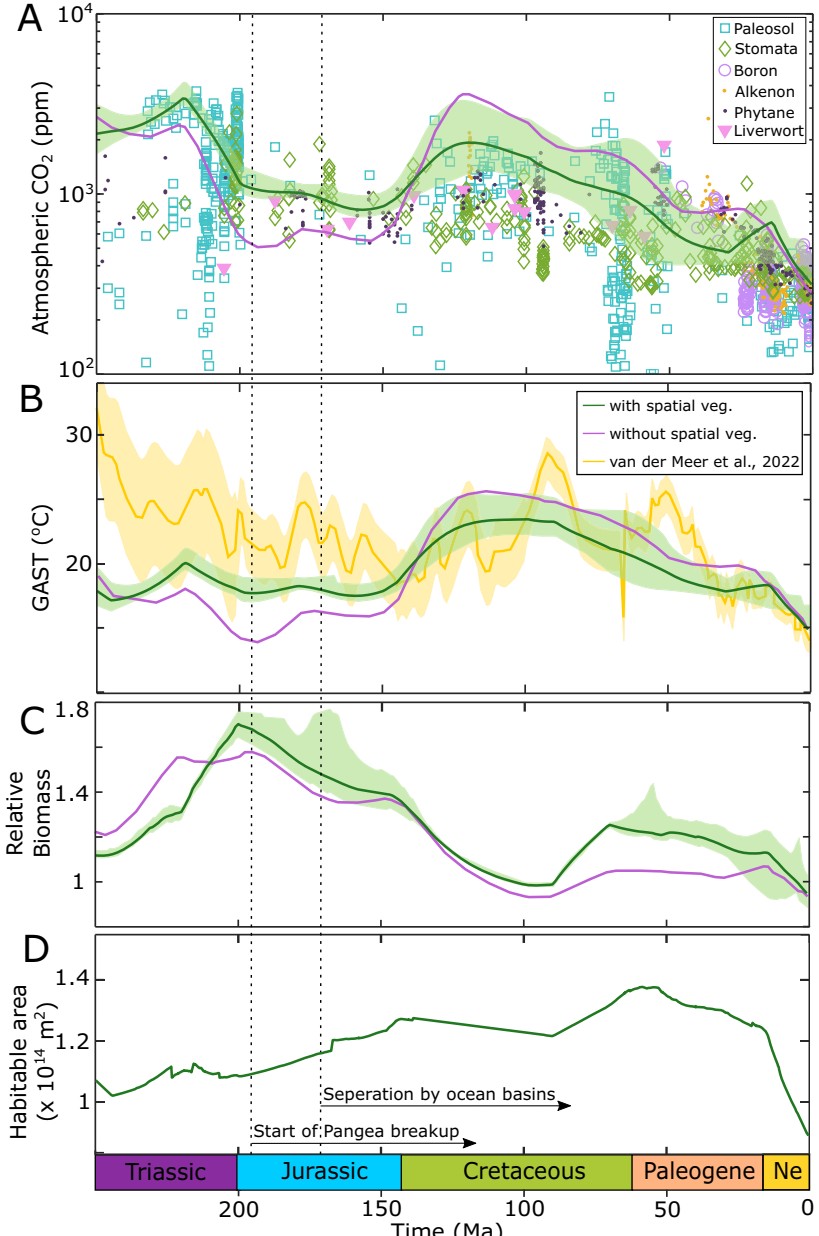

**Fig. 3 | Mesozoic-Cenozoic time dependent model solutions.** Comparison between SCION (purple line), SCION-FLORA (green line) model outputs and proxy records of **A** atmospheric $CO_2$ predicted from $\delta^{13}C$ of paleosols, alkenones, phytane and liverworts, stomatal ratio and boron isotopes in foraminifera (data taken from refs. [22], [43]), **B** global average surface temperature (GAST) (yellow line;[44]), **C** relative global vegetation biomass, and **D** total area of continents with plants present. SCION uses a box model approach for vegetation and therefore does not use habitable area to calculate vegetation. Sensitivity analysis results fall within the green shaded areas. Dotted lines indicate the start of the breakup of Pangaea and the presence of narrow ocean basins separating Laurasia, and Eastern and Western Gondwana[45]. Ma million years ago, Ne Neogene.

climate during model spin-up in which the vegetation was eliminated and $CO_2$ reached high levels which caused the model to crash. Similar dynamics have been seen in other simpler models[33]. Each run varied in their degassing rates (upper and lower boundaries from ref. [16]), distribution of lithologies between 'basaltic' and 'granitic' (±20%), and isotopic fraction factors for carbon (for photosynthesis, ±5‰) and sulfur (for microbial sulfate reduction, ±10‰), to create an uncertainty range.

### Model equations
The biotic enhancement of silicate weathering, $f_{biota}$, consists of a linear relationship with FLORA NPP and a constant abiotic factor

($f_{minbiota} = 0.32$) using the relative atmospheric $CO_2$, $RCO_2$:

$$f_{biota} = 0.0005(NPP) + f_{minbiota} \cdot RCO_2{}^r \qquad (1)$$

This embodies the assumption that higher NPP will lead to a greater degree of biotic enhancement of silicate weathering–an approximately linear relationship between NPP and silicate weathering is apparent from global catchment[17,34], biomass and NPP data[35,36] (Figure S5). The large range of potential vegetation enhanced weathering rates[37] presents a challenge to forming a single relationship between biomass and weathering. Here $f_{biota}$ increases by a maximum 6-fold between bare ground and maximum NPP grid cells, roughly

consistent with the 4- to 7-fold enhancement factors assumed in global biogeochemical models[8,10,11]. The GEOCARB models use a function ($RCO_2{}^r$) to represent pH-driven abiotic weathering rates in the absence of a land biosphere, where $a$ is between 0.25 and 0.5[38,39]. We adopt this formulation here for the uncolonized grid cells in SCION-FLORA, and choose $r = 0.25$ as the more conservative estimate, acknowledging that NPP-weathering relationships require further study.

Within FLORA, biomass death, $B_{turnover}$, and biomass ($B$) are updated to include the consequences of increased atmospheric oxygen:

$$B_{turnover} = \min\bigl(\max(0.0092 \cdot (100 \cdot O_{2mr} - 10), 0.08), 0.2\bigr) \quad (2)$$

To account for the increased probability of fire at higher oxygen concentrations[40], we consider that as oxygen increases from 17%, turnover increases until it reaches a maximum of 20% (i.e., double the turnover at present-day oxygen levels). Biomass itself is not carried over to the next SCION timestep and is re-seeded in FLORA to run to stability. The relative total vegetation at steady state in FLORA is used as a factor to calculate terrestrial $C_{org}$ burial rates, which also impacts terrestrial P burial and reduces P influxes to the ocean in SCION.

Leaf respiration, $s$, uses daylight hours, $h$ to determine an overall photosynthesis rate:

$$s = \left(\frac{24}{h}\right)a \quad (3)$$

Due to the long timescales and lack of seasonality, we assume that the average sunlight hours is 12 h per day across all latitudes per year, hence $h = 12$. Two parameters deal with two different aspects of light limitation: time of exposure and intensity. The change in light intensity impacts productivity by limiting photosynthesis as Rubisco activity is determined by intracellular $CO_2$ and $pO_2$ levels, and $s$. The latitudinal distribution of insolation accounts for the strength of light passing through the atmosphere and the decreased luminosity in the earlier Phanerozoic relative to the present day (roughly a 5% decrease)[41].

## Data availability
All data used to generate biomass in this study has been deposited in the Supplementary Code file. It can also be accessed via GitHub Khushgrg/SCI-FI (https://doi.org/10.5281/zenodo.10610687[42]).

## Code availability
The SCION model code can be accessed at https://github.com/bjwmills/SCION. The FLORA model code can be accessed at https://github.com/KhushGrg/FLORA. The full SCION-FLORA model code used here (and required datasets) can be accessed at Khushgrg/SCI-FI (https://doi.org/10.5281/zenodo.10610687[42]).

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

## Acknowledgements

This work was supported by the Natural Environment Research Council [grant number NE/S009663/1]. K.J.F. is supported by a BBSRC Translational Fellowship (BB/M026825/1) and ERC CoG "MYCOREV" (865225). S.A.B. was supported by the Natural Environment Research Council [grant number NE/M019497/1].

## Author contributions

This paper was conceptualised by K.G., B.J.W.M., and K.J.F. Funding was acquired by B.J.W.M., K.J.F., S.A.B., and S.W.P. K.G. and B.J.W.M. did the methodology. The software, validation, visualisation, and writing of the original draft was done by K.G., supervised by K.J.F. and B.J.W.M. The draft was reviewed and edited by all authors. The project administration was done by B.J.W.M.

## Competing interests

The authors declare no competing interests.
