## [Peer Review File · Nature Communications]

Geographic range of plants drives long-term climate changeReviewer #1 (Remarks to the Author):

MANUSCRIPT REVIEW: Geographic range of plants drives long-term climate change

The paper shows a simulation of biomass variation, CO₂ concentration in the atmosphere, surface temperature and the geographical distribution of vegetation for the planet since the Triassic period (220 million years ago). In this simulation, the authors use the FLORA model (derived from LP-DGVM) coupled to SCION (for weathering).

An interesting aspect of the result and is the most relevant point of this manuscript was to present Pangea, a supercontinent that had a large semi-arid area in its interior due to the lack of moisture transport from the oceans; and so, as a consequence, the results point to less biomass, a higher concentration of CO₂ in the atmosphere (less sink), higher temperatures and less runoff. And on the other hand, in the course of time, in the Jurassic period, when Pangea splits, there is greater circulation of moisture from the oceans over the terrestrial parts, therefore, more vegetation, biomass and removal of atmospheric CO₂ (sink).

Also interesting are the comparisons between SCION-FLORA and SCION alone, which allows us to think that the coupling had a more comprehensive result.

The manuscript has relevance and originality, however, some important processes of conducting the simulations and results need to be explained and made clearer before it can be published.

CRITICAL POINTS, DOUBTS AND/OR QUESTIONS:

1. One of the points that is not clear is that the authors describe that one of the parameters used to estimate local rates of photosynthesis is the latitudinal distribution of solar radiation. But on the other hand, they also considered that the entire surface has an exposure of 12 hours of sunlight per day. If so, I don't understand why they used the latitudinal distribution, if the entire surface receives 12 hours of sunlight per day?

The other parameters for estimating photosynthesis, in addition to solar radiation, are global O₂ and CO₂ levels, temperature, and runoff.

1 I believe that if there is runoff, it is because there is precipitation (why don't the authors cite precipitation anywhere in the manuscript?). However, they cite the transport of moisture from the oceans.

2. Between lines 81 and 89, the authors describe how the SCION-FLORA coupling is performed and how the rounds are performed. Since every ~100,000 years approximately, there is a new biomass seeding for each gridcell of the surface performed in the SCION model, and this is taken to the FLORA model which seeks a "steady" state when the biomass change between one step and another is less than 1%, and the NPP is then used to calculate the rates of intensification of silicate weathering in the next timestep ($k+1$). Is this what the authors meant? If that's right, I think it needs to be clearer to the reader, as in the text, as in the figure. Are there two loops (one for FLORA and one for SCION), where the time difference between the two models is only $k+1$? What is the k ? 100,000 years, 1 year, or one day? The number 852 successful runs?

3. In Figure 3, the authors place the last geological period as Ne and do not explain anywhere (not even in the caption of the figure) what the abbreviation for the Neogene period is about.

Also in Figure 3, the authors put results (I believe they are from SCION) of paleosol, boron, alkenon, phytane, livewort, and the stomata of vegetation, but there is in the text any citation or interpretation of these results for the reader. I was surprised that the authors put all this in the picture, and didn't devote a paragraph to commenting/interpreting these results? The increase of stomata what is it: increase in the number of stomata in the vegetation, or increase in stomatal function? There is an increase in paleosol at the end of the Triassic period when biomass is rising, and CO₂ decreases upon absorption (sink). How do the authors interpret this result?

Also in Figure 3, which is the most important synthesis result of the manuscript, there is an inflection point between the Paleogene and Neogene periods, which occurs for atmospheric CO₂ (more intense), temperature, biomass and habitable area. How do the authors also explain this result?

They are data and details that cannot be omitted and that if interpreted (commented) enrich the manuscript making it more attractive to the reader.

Reviewer #2 (Remarks to the Author):

The Authors present an innovative approach to incorporating core elements of a dynamic vegetation model into a relatively well-time-resolved, global climate-biogeochemical modeling framework. This results in an estimate of global vegetative biomass for the last 220 million years, and the concomitant alteration of geochemistry, atmosphere, and climate variables. This represents a valuable step in the integration of biological and geochemical influences in models of the global carbon cycle, and therefore understanding of Earth system functioning. Both the results and framework will be valuable to all biogeoscientific fields for which vegetative biomass and biogeography, and chemical terrestrial-marine connections are relevant.

The results could be more clearly framed in terms of the limitations of the modeling approach, which are considerable by both design and necessity, described more adequately in previous publications, and do not detract from the value of the work. Nonetheless, they are necessary to frame the work and must be treated with more detail. For example, the previous FLORA publication stated, "The intention here is to understand the biomass potential of past climates based on fundamental photosynthetic processes and parameters." This does not need to be repeated throughout but does need to be included as an interpretive lens when introducing and discussing the results (e.g., presenting Figure 2 as potential biogeography).

The description of the Methods left some ambiguity and made results challenging to interpret. This was particularly an issue concerning the details of the FLORA- coupling and the sensitivity analysis.

As it is the crucial variable connecting the biosphere to climate, the paper is missing one or two high-impact comparisons to established paleontological thought on biogeography.

Finally, the current manuscript is missing some detail on how the model processes produced the reported results and, as I alluded to before, in what way model assumptions bias the results.

I have left detailed comments in an attached Word document.

All of the issues mentioned could be addressed with relatively minor edits to the current main text (possibly including tweaking the title, which is very general), the addition of additional methods and model caveats sections to the supplement, a discussion of the detailed model results in the supplement, an improvement of Figure 1 to better represent the information transfer to and from FLORA and the other components of SCION, and modification to Figure 2 to include the SCION ensemble results.

I recommend that the Editor accept this manuscript once these issues are addressed.

In their review of the first version of this manuscript, reviewer #2 added some comments to the manuscript file. These comments were forwarded to the authors, who replied as included in this Peer Review File.

Reviewer #3 (Remarks to the Author):

This manuscript couples a vegetation model (FLORA) to a spatially resolved long-term Earth System model (SCION) to show that vegetation would have been sparse in the dry interior of Pangaea and, as a result, plant-assisted CO₂ fixation would be less than previously thought. The study finds that the difference between the new FLORA-coupled SCION model and the old SCION model give rise to atmospheric CO₂ levels up to double previous predictions. Finally, the study finds that the model fits geological data better (ice line and pCO₂ proxy data) and conclude that the geographical range of plants have exerted first-order control on long-term climate change.

I note that the manuscript follows after a previous publication in Nature Communications (Gurung et al., 2022) showing that climate changes due to changing continental configurations opened windows for geographical expansion of land plants in the Ordovician and Jurassic-Paleogene. That is, Climate  Vegetation. The submitted manuscript aims to make the opposite link: Vegetation -> Climate.

First of all, the main conclusion that vegetation coverage influences atmospheric CO₂ levels, thereby climate, is expected. Implicitly, plant coverage has been assumed in previous COPSE models and this has driven climate change via changing atmospheric CO₂ levels.

That said, it is a valid endeavor to try and test whether plant coverage affected climate. The study seeks to compare a model that predicts plant coverage (FLORA+SCION) with one that does not (SCION). It is not clear how the new model predicts plant coverage. Previous work (not cited here) have already suggested that the geographical range depends on water availability in the continental interiors (e.g. Maher & chamberlain 2014; Ibarra et al. 2019). Given that SCION model is not a Global Circulation Model (GCM), it lacks the ability to predict water availability in the continental interiors and therefore cannot predict plant coverage very well. It seems that SCION employs a highly idealized parameterization between runoff and vegetation, which ignores evapotranspiration (one of the key factors thought to affect water availability on continents). Therefore, I would think that FLORA-SCION model has a poor chance of predicting the geographical range of plants.

I think it is difficult to convey how this result is new. I'm not convinced that that the methods used are adequate to make the conclusion (plant coverage have dictated CO₂ levels). I do see that that the predicted CO₂ curve looks to be closer to average value of the pCO₂ proxy data than the previous SCION model, but I'm worried that it is not a significantly better fit given the uncertainty of the available proxy records.

Further, advanced vegetation models (LPJ-GUESS) have demonstrated that positive feedbacks exist on plant coverage (vegetation increase water availability inland via evapotranspiration) in ways that can only be predicted if vegetation models are coupled with GCMs (Lu et al. GRL 2019). Therefore, this study ignores some existing knowledge in the field, which makes it look like an (perhaps important) incremental improvement to a simplified long-term Earth system model.

Details:

L. 129. Missing .

L. 135 There is little difference in the Cenozoic Neogene results.

L. 141. How is "Limited water circulation during the time of Pangaea" calculated? Or is this an implicit assumption?

L165. Jargon/ill-defined term. What is "biogeochemistry" referring to here: "However, the global effect of plants on biogeochemistry [...]"

REVIEWER COMMENTS

Reviewer #1 (Remarks to the Author):

MANUSCRIPT REVIEW: Geographic range of plants drives long-term climate change

The paper shows a simulation of biomass variation, CO₂ concentration in the atmosphere, surface temperature and the geographical distribution of vegetation for the planet since the Triassic period (220 million years ago). In this simulation, the authors use the FLORA model (derived from LP-DGVM) coupled to SCION (for weathering).

An interesting aspect of the result and is the most relevant point of this manuscript was to present Pangea, a supercontinent that had a large semi-arid area in its interior due to the lack of moisture transport from the oceans; and so, as a consequence, the results point to less biomass, a higher concentration of CO₂ in the atmosphere (less sink), higher temperatures and less runoff. And on the other hand, in the course of time, in the Jurassic period, when Pangea splits, there is greater circulation of moisture from the oceans over the terrestrial parts, therefore, more vegetation, biomass and removal of atmospheric CO₂ (sink).

Also interesting are the comparisons between SCION-FLORA and SCION alone, which allows us to think that the coupling had a more comprehensive result.

The manuscript has relevance and originality, however, some important processes of conducting the simulations and results need to be explained and made clearer before it can be published.

We thank Reviewer 1 for their encouraging comments. We address their concerns below

CRITICAL POINTS, DOUBTS AND/OR QUESTIONS:

1. One of the points that is not clear is that the authors describe that one of the parameters used to estimate local rates of photosynthesis is the latitudinal distribution of solar radiation. But on the other hand, they also considered that the entire surface has an exposure of 12 hours of sunlight per day. If so, I don't understand why they used the latitudinal distribution, if the entire surface receives 12 hours of sunlight per day?

The two parameters deal with two different aspects of light limitation: time of exposure and intensity. The latitudinal distribution of insolation accounts for the strength of light passing through the atmosphere which differs due to Earth's spherical shape and also the decrease of luminosity over the Phanerozoic (Kasting 1989). The change in light intensity is assumed to impact productivity by limiting photosynthesis. The exposure to sunlight however deals with the amount of time that photosynthesis is active. Rubisco activity is determined by intracellular CO₂ and pO₂ levels and the additional daylight term (sigma) decreases the overall photosynthesis rate. And as the model deals in such large timescales it does not resolve seasonal cycles where day length can differ substantially by latitude. We have therefore set sunlight hours as 12 hours per day, which is the approximate average for the

year across all latitudes. We have articulated this justification for our approach more clearly in the methods sections (Lines 275-282).

1. I believe that if there is runoff, it is because there is precipitation (why don't the authors cite precipitation anywhere in the manuscript?). However, they cite the transport of moisture from the oceans.

The runoff parameter is taken from a set of FOAM climate model runs where, unfortunately, the precipitation parameter was not saved. Although runoff was calculated using precipitation, which takes into account the movement of land and water transport from the oceans, we do not have precipitation data over the Phanerozoic to be used in the SCION model. Fortunately, the FLORA model has previously been tested against global biomass and performs reasonably well when using runoff as a forcing. We have noted this in the text, and that future work could potentially be improved by using a wider range of outputs from climate models (Lines 111-117).

2. Between lines 81 and 89, the authors describe how the SCION-FLORA coupling is performed and how the rounds are performed. Since every $\sim 100,000$ years approximately, there is a new biomass seeding for each gridcell of the surface performed in the SCION model, and this is taken to the FLORA model which seeks a "steady" state when the biomass change between one step and another is less than 1%, and the NPP is then used to calculate the rates of intensification of silicate weathering in the next timestep ($k+1$). Is this what the authors meant? If that's right, I think it needs to be clearer to the reader, as in the text, as in the figure. Are there two loops (one for FLORA and one for SCION), where the time difference between the two models is only $k+1$? What is the k ? 100,000 years, 1 year, or one day? The number 852 successful runs?

Yes, the reviewer is correct in their interpretation. K is indeed the SCION step of $\sim 100,000$ years. We apologise for the very short explanation of the model. We have added a longer explanation of how the model works in Lines 82-94 and updated Figure 1 to better represent the exact flowchart of the model and improve clarity.

A brief explanation of the sensitivity analysis is added to the Methods section (Lines 246-252). We ran the model 1000 times using a range of parameter uncertainties. This resulted in 852 runs completing (hence giving us the error range). 148 runs did not complete as the model was not able to complete the spin-up phase under some more extreme parameter choices.

3. In Figure 3, the authors place the last geological period as Ne and do not explain anywhere (not even in the caption of the figure) what the abbreviation for the Neogene period is about.

Our apologies. We have added 'Ne:Neogene' to the figure legend.

Also in Figure 3, the authors put results (I believe they are from SCION) of paleosol, boron, alkenon, phytane, livewort, and the stomata of vegetation, but there is in the text any

citation or interpretation of these results for the reader. I was surprised that the authors put all this in the picture, and didn't devote a paragraph to commenting/interpreting these results? The increase of stomata what is it: increase in the number of stomata in the vegetation, or increase in stomatal function? There is an increase in paleosol at the end of the Triassic period when biomass is rising, and CO₂ decreases upon absorption (sink). How do the authors interpret this result?

We regret our lack of explanation here. These are all proxies for atmospheric CO₂ concentration to which we compare the model CO₂ prediction. We have now added some text on this. (see Lines 142-149).

Also in Figure 3, which is the most important synthesis result of the manuscript, there is an inflection point between the Paleogene and Neogene periods, which occurs for atmospheric CO₂ (more intense), temperature, biomass and habitable area. How do the authors also explain this result?

This is a good question. Modelled CO₂ increases by ~200 ppm across the Paleogene-Neogene boundary (30 - 15 Ma). This is partly due to the small decline in total vegetation biomass (1.2 to 1.17 relative vegetation biomass). But more importantly, as our paper suggests, it is to do with the spatial location of vegetation. As we go from 30 to 15 Ma, we lose weathering enhancement in Australia and South East Asia and some parts of Africa, which ultimately increases CO₂. The effects of this can also be seen in Figure S3C as the C_{org} flux has a steeper decline between 50-15Ma compared to the original SCION results. We have added an explanation and supplementary figure to the paper to strengthen our conclusions here (Lines 195-200).

They are data and details that cannot be omitted and that if interpreted (commented) enrich the manuscript making it more attractive to the reader.

We thank the reviewer for these constructive points which have improved the paper.

Reviewer #2 (Remarks to the Author):

The Authors present an innovative approach to incorporating core elements of a dynamic vegetation model into a relatively well-time-resolved, global climate-biogeochemical modeling framework. This results in an estimate of global vegetative biomass for the last 220 million years, and the concomitant alteration of geochemistry, atmosphere, and climate variables. This represents a valuable step in the integration of biological and geochemical influences in models of the global carbon cycle, and therefore understanding of Earth system functioning. Both the results and framework will be valuable to all biogeoscientific fields for which vegetative biomass and biogeography, and chemical terrestrial-marine connections are relevant.

The results could be more clearly framed in terms of the limitations of the modeling

approach, which are considerable by both design and necessity, described more adequately in previous publications, and do not detract from the value of the work. Nonetheless, they are necessary to frame the work and must be treated with more detail. For example, the previous FLORA publication stated, "The intention here is to understand the biomass potential of past climates based on fundamental photosynthetic processes and parameters." This does not need to be repeated throughout but does need to be included as an interpretive lens when introducing and discussing the results (e.g., presenting Figure 2 as potential biogeography).

The description of the Methods left some ambiguity and made results challenging to interpret. This was particularly an issue concerning the details of the FLORA- coupling and the sensitivity analysis.

As it is the crucial variable connecting the biosphere to climate, the paper is missing one or two high-impact comparisons to established paleontological thought on biogeography.

Finally, the current manuscript is missing some detail on how the model processes produced the reported results and, as I alluded to before, in what way model assumptions bias the results.

I have left detailed comments in an attached Word document.

We thank the reviewer for their careful remarks and summary and have pasted these detailed comments below to answer them in full. As the points above are covered in more detail below we do not answer them here.

All of the issues mentioned could be addressed with relatively minor edits to the current main text (possibly including tweaking the title, which is very general), the addition of additional methods and model caveats sections to the supplement, a discussion of the detailed model results in the supplement, an improvement of Figure 1 to better represent the information transfer to and from FLORA and the other components of SCION, and modification to Figure 2 to include the SCION ensemble results.

I recommend that the Editor accept this manuscript once these issues are addressed.

We thank the reviewer for such a detailed review of the paper and for reading our prior work related to this. Your critical discussion of this paper has definitely improved the work for which we are very grateful. We will leave thoughts on the title to the editor if they accept the paper, and endeavour to respond to all of the reviewer's points below.

Comments transferred from Word Document:

1. I'm pretty sure I know what you are trying to say here, but people who are not used to thinking about simulated vs actual vegetation distribution might like a little more clarity here, especially as this is a key problem in the field.

Yes, that is true. "Simulated vegetation" as you specified has been added to the abstract (Line 17).

2. might specify time for parallelism with the next sentence.

Added time frame “Triassic-Jurassic” for context (Line 22).

3. Does atm. CO₂, per se, stabilize climate? I didn’t get that from 3,4.

We have removed the sentence to make the point short and clear. CO₂ is a component of the mechanisms that stabilise climate.

4. burning isn’t so clearly just a source. check out (10.1111/gcb.12985, 10.1111/gcb.13603). could rearrange to distinguish fire emissions from PyC, or use a simpler example.

We now use ‘pyrogenic carbon from biomass burning’ (Line 40).

5. philosophical side note. some paleobotanists prickle at the phrase “land plants,” as redundant.

Understood, but the convention in the fields we build on has been to use ‘land plants’ to emphasise that they contribute to land surface processes like continental weathering, which is very important in this work, so we would like to retain the phrase this time but will consider trying to move away from it.

6. a bit anthropomorphic? this seems to connect back to the ‘stabilisation’ language above, which I also found a little unclear.

Altered to “climate regulation through the Mesozoic to Cenozoic” (Line 61).

7. I don’t understand what “consistent” the second phrase means here. The steady state thing is explained, but what comparison between FLORA and SCION variables is used as a stopping criterion (etc.)?

We have changed the explanation of the model framework to make it a clearer (Lines 82-94).

8. I’m a bit confused here. Feedback in what? Do the variables passed to SCION alter vegetation growth in the subsequent run of FLORA? Also, you establish vegetation at the kth time, and then use the weathering enhancement, Corg burial and NPP from that time for the subsequent time, ~100ky later? If this is the case, a discussion of the timescale of biome turnover, colonization, desertification etc. is warranted in a ‘caveats’ section.

We have added ‘feedback between vegetation and climate’ within the model explanation above. And yes this is a good point, we prescribe the same biosphere for ~100kyrs of long-term climate variation. We now note that many changes in the biosphere could take place on shorter timescales, so our results should be seen only as an estimation of the potential long-term outcomes when shorter term events are removed (Lines 111-117; 201-217).

9. which variables are going from FLORA to the GCM component of SCION, if any? how can that be possible at each SCION timestep, if the GCM is only run at coarser timesteps? if you are feeding variables back into the GCM then why not hydrological cycle variables too? see Ibarra et al. 2019 (10.2475/01.2019.01)

Apologies for the lack of information here. We have expanded the section to contain a more detailed explanation of the entire model. Nothing is passed to the GCM from the vegetation model as we rely on previously-computed GCM runs to build an emulator - otherwise the entire model would not be able to run over deep time (Lines 64-65). The time period and CO2 level is what dictates the climate fields within the emulator. Lack of changes in GCM hydrology in response to plants is a key limitation and we now cite the Ibarra paper here (Line 204). Solving this problem with present computing resources is not currently possible in our opinion although we suggest a way forwards which could involve running climate models with and without plants, perhaps combined with machine learning approaches.

10. Depending on the answers to the above two questions, a slightly more detailed schematic would be helpful. Specifically, I'd like to be able to relate this to the schematic of SCION from the 2021 Mills et al. paper.

In addition to the above explanation, we've also modified Fig 1 to include processes involved in the SCION-FLORA interaction.

11. what does this mean?

It means that the distribution of the biomass is not prescribed. We have changed 'emergent' to 'modelled' (Line 104).

12. In the previous FLORA paper you qualified this as 'potential' biomass. I would think that should carry over, and also be applied to 'vegetation' throughout. Actually, I think the later is more appropriate since you have eliminated allocation and don't consider differences in the resistance to decomposition of different biomass types (e.g., secondary xylem vs leaf mesophyll).

Previously we used 'potential biomass' because we were running the model for time periods where we knew plants had not evolved, or were much more primitive than our equations allowed for. So we feel using the same term here would be inconsistent with our previous work. But we understand the reviewer's point so will use the 'modelled vegetation' etc. to be clear that this is still a simplified approach.

13. it is worth mentioning the angiosperm innovations here and the possibility that step changes in reproductive and water transport traits could bias your results both in terms of biomass and range, particularly where you compare the mesozoic to the cenozoic below.

A very good point and we now mention this further in the discussion (Lines 201-211)

14. FLORA is the first model I've seen use runoff as plant available water. I looked at the previous paper a bit, but did not see a justification for this. Apologies if I missed something, but I think it's really necessary for comparisons to other work to include the reasoning behind this choice. Even if it's due to data availability, a discussion of the ways using runoff could change your results would be really helpful. If I'm reading the FOAM 1.5 docs correctly the soil depth is pretty shallow (15 cm). This means you have a threshold for starting plant growth at the soil water capacity. Soil evaporation is also taken before runoff, this means that you will underpredict vegetation in many places around that threshold. On the other hand, because you are using runoff, places that fall along simulated river routes may have much more plant available water than if you used actual soil moisture. Alongside this complex bias, it also seems that FLORA produces an outer bound for vegetation biogeography in terms of temperature and radiation. I'd be really interested to see the Authors' thoughts on how these model processes affect their results.

FLORA uses runoff instead of precipitation or soil moisture simply because the FOAM climate model dataset used within the broader SCION biogeochemical model only contains runoff information for the Phanerozoic. The model runs were initially carried out to look at chemical weathering, and we are in the process of producing new climate model runs where we can look at other important variables like soil moisture, precipitation and evaporation. We have included a brief discussion in the revision and speculated how this could impact our results, although as we showed in our previous paper on FLORA, it generates a reasonable prediction of present day biomass when forced with runoff data.

15. I don't think this is a useful statement. The biome-specific statement below, is more useful. To my eyes, there are large discrepancies in certain biomes. This is not unexpected considering the the representation of plants and water availability. You could couch this section in a specific comparison to actual biomass data like that in ForC (10.1002/ecy.2229), or the ESA dataset (eo4sd-forest.info/global-biomass/). Of course you would have to aggregate the higher resolution data, but this would make an interesting figure. And it would provide a very valuable touchstone for your analysis that most can't access, exactly because they are unable to such broad scales in time in space. You might also consider introducing an error metric to make this quantitative.

We have revised this text to more carefully state the discrepancies between SCION-FLORA biomass and the observed real-world biomass. We have also been clearer to note that a high-resolution comparison of FLORA to present day biomass is already explored in our previous open-access paper on the model. Additional analysis can be found in the supplementary.

16. This is certainly a consideration, but there are many other reasons for differences from modern

Yes, there are multiple reasons for the mismatch between predicted and modelled data. We have added more reasons for this throughout the revised paper and in the supplementary.

17. what do the grey areas in the first and third columns mean? What does 'the shifting locus of silicate weathering' mean? generally, this figure-caption is not interpretable on its own. if the grey areas are places where the vegetation model failed to reach steady-state, what does that mean for the overall analysis. it certainly seems to contradict your currently over-general statement in the results-discussion section regarding agreement between simulated and actual modern biogeography.

Grey areas indicate no biomass present due to no runoff or temperatures $< -10^{\circ}\text{C}$. We have added this to the Figure 2 caption. The 'Shifting locus of silicate weathering' expresses the changing locations of weathering from following the breakup of Pangea. A brief explanation is also included with Figure 2 now.

18. this assumption, critical to the results, has been challenged elsewhere. specifically, that weathering rates per se are not increased by plants (D'Antonio 2020, 10.1130/G46776.1) the counterpoints to your assumption should be mentioned somewhere. and interestingly, it seems like this model is poised to test the assertions of the above paper.

We intended 'biotic enhancement' to mean an increased feedback strength, not an overall increase in weathering rates (which are also controlled by a number of other factors). So we are in agreement with the D'antonio paper. We have made this clearer in the revision.

19. a number of people are going to look at your biogeography for 200 Mya and wonder where the Central Atlantic Magmatic Province is. they would then think that the CAMP would knock a big chunk out of your maximum biomass in figure 3. it would be worth mentioning explicitly that this approach is not trying to capture perturbations like that at this stage. i think there is a statement along these lines in the SCION paper. since you talk about the Triassic a fair amount here, and highlight a timepoint at the T-J, this is an opportunity to clarify current limitations while pointing to future work, using a well studied example. of course, you don't want to get into much detail (trouble). just the highlights in a very brief form.

This is a good point. We do not model any LIP events and have added a line to clarify this (Line 212-217).

20. water vapor transport?

Yes – we have now written 'moisture'

21. Would be good to give information in the methods about what this means (i.e., how and why the model 'fails', model sensitivity to what variables). Maybe I missed it, but I actually don't see any mention of this in the Methods.

As this was mentioned by a previous reviewer, a more detailed explanation of the sensitivity analysis has been added to 'Methods' (Lines 246-252). The model fails to spin up (e.g. the vegetation all dies and CO² levels become too high for the emulator) if the chosen parameters are too far from the model starting state.

22. cliffhanger..

Apologies about that. It now includes a conclusion. "Comparison of model results to other climate proxies and overall fluxes can be found in Figure S3."

23. see comments about the CAMP and about river routing and water availability.

LIP forcing and the use of runoff is mentioned now in limitations.

24. this all depends heavily on the simple relationship between NPP and weathering.

Yes, we have made that clearer in the limitations section.

25. Because the sensitivity analysis is only applied to the new model, and also not actually described in the text, it is difficult to compare the models or the statements about agreement between the models and proxy data based on this figure.

Does this follow the same ensemble paradigm as in the SCION publication? Why isn't the SCION ensemble area presented?

Text to describe the sensitivity analysis has been added. The SCION ensemble varies approximately symmetrically around the median value thus we include only the median line from that model to help reduce visual clutter.

26. why is this presented as % area? wouldn't the absolute area be more relevant to a discussion of weathering impact on climate?

Thank you for this suggestion. We have changed from % to absolute area in Fig 3.

27. this is a bit ambiguous, as with the previous reference to ocean-land water relations. could use a reference and clarification.

Reference and details added. (Lines 185-187).

28. is this just SCION? why not specify and avoid ambiguity?

Apologies for the ambiguity. Specified now to only SCION.

29. these are the only two uses of these words in the paper. use key words that link back to specific discussion points above. it really helps readers.

Good point. Changed to "... the importance of the inclusion of geographical spread of plants on weathering enhancement rates..." (Line 224)

30. this is the initial condition at every grid cell? what does the last phrase mean? why not do another end-member experiment with a very low C initial condition? wouldn't that represent re-establishment?

Yes, every grid cell is seeded with biomass every iteration of SCION so every~100kya. Apologies on the typo, it is 2.5×10^4 and not 10^{14} . 're-establishment' means that each gridcell can generate biomass even if it became barren in the previous timestep. We have clarified this. Changing the initial biomass to a lower value does not influence the results as the vegetation model quickly moves away from this value towards steady state.

31. what happens to the C in biomass after death? does this influence the Corg rates in SCION?

Corg burial rates are based on total vegetation present only. This is now explained now in Methods. Lines 271-273.

Reviewer #3 (Remarks to the Author):

This manuscript couples a vegetation model (FLORA) to a spatially resolved long-term Earth System model (SCION) to show that vegetation would have been sparse in the dry interior of Pangaea and, as a result, plant-assisted CO₂ fixation would be less than previously thought. The study finds that the difference between the new FLORA-coupled SCION model and the old SCION model give rise to atmospheric CO₂ levels up to double previous predictions. Finally, the study finds that the model fits geological data better (ice line and pCO₂ proxy data) and conclude that the geographical range of plants have exerted first-order control on long-term climate change.

I note that the manuscript follows after a previous publication in Nature Communications (Gurung et al., 2022) showing that climate changes due to changing continental configurations opened windows for geographical expansion of land plants in the Ordovician and Jurassic-Paleogene. That is, Climate -

-> Vegetation. The submitted manuscript aims to make the opposite link: Vegetation  Climate.

This is a concise and clear summary of the conclusions and novelty our work and we thank the reviewer for reading it in detail

First of all, the main conclusion that vegetation coverage influences atmospheric CO₂ levels, thereby climate, is expected. Implicitly, plant coverage has been assumed in previous COPSE models and this has driven climate change via changing atmospheric CO₂ levels.

We agree that some relationship is expected, but we found the relationship to be complex. We have been clearer to note that previous models like COPSE considered only global biomass, which as we show here, may not trend in the same direction as plant effects on weathering. (Line 164)

That said, it is a valid endeavor to try and test whether plant coverage affected climate. The study seeks to compare a model that predicts plant coverage (FLORA+SCION) with one that does not (SCION). It is not clear how the new model predicts plant coverage. Previous work (not cited here) have already suggested that the geographical range depends on water availability in the continental interiors (e.g. Maher & chamberlain 2014; Ibarra et al. 2019).

We have been clearer to note that the FLORA vegetation model is already published open access and we have include more detail on its function and coupling to SCION in the revision. We agree that previous work has suggested that water availability will impact plant coverage and have now added the suggested citations.

Given that SCION model is not a Global Circulation Model (GCM), it lacks the ability to predict water availability in the continental interiors and therefore cannot predict plant coverage very well.

We have been clearer in the revision that SCION does actually include a steady state GCM emulator and does predict continental hydrology and temperature as spatial fields, which allow us to estimate plant coverage that shows a similar pattern to data for the present day.

It seems that SCION employs a highly idealized parameterization between runoff and vegetation, which ignores evapotranspiration (one of the key factors thought to affect water availability on continents). Therefore, I would think that FLORA-SCION model has a poor chance of predicting the geographical range of plants.

While FLORA does not directly parameterise vegetation entirely from runoff, it does use runoff to inform water stress, which impacts plant growth. This is a key limitation and runoff is used because of the limited available fields in our climate emulator. However we have shown in Gurung et al. (2022) that FLORA predictions that use runoff in this way give a reasonable reconstruction of global present day biomass. We have made this limitation much clearer in the revision and have included a comparison of SCION-FLORA biomass to present day data, which shows it can reproduce the key trends over latitudes and longitudes. We are working on a new climate emulator which will help solve these issues, but performing such a large number of GCM simulations takes several years.

I think it is difficult to convey how this result is new.

We are now clearer in stating the novelty of the work. Following the reviewer's clear summary in their introduction, we note that this is the first paper to our knowledge to explore quantitatively the bi-directional feedbacks between the spatial distribution of plants and long-term climate change over the Mesozoic and Cenozoic (Line 223). As far as we understand it, nobody has previously coupled a vegetation model to a 3D Earth system model dynamically over deep time (e.g. >> 1 Myr).

I'm not convinced that that the methods used are adequate to make the conclusion (plant coverage have dictated CO₂ levels). I do see that that the predicted CO₂ curve looks to be closer to average value of the pCO₂ proxy data than the previous SCION model, but I'm worried that it is not a significantly better fit given the uncertainty of the available proxy records.

We now use a distance metric to calculate the closeness of the curves, which helps us be more precise in our conclusions. We have also been clearer that our main comparison is with the paleotemperature curve rather than the CO₂ curve due to the large uncertainty of the pCO₂ proxies (Lines 154-156).

Further, advanced vegetation models (LPJ-GUESS) have demonstrated that positive feedbacks exist on plant coverage (vegetation increase water availability inland via evapotranspiration) in ways that can only be predicted if vegetation models are coupled with GCMs (Lu et al. GRL 2019). Therefore, this study ignores some existing knowledge in the field, which makes it look like an (perhaps important) incremental improvement to a simplified long-term Earth system model.

We agree that a direct coupling between the vegetation model and a GCM would be preferable, but that sort of analysis can only be done over geologically-short timeframes (e.g. 2000 years in Lu et al.) due to computational constraints – which is why evapotranspiration feedback is ignored in our work. We have been clear to note this limitation in the revision and suggest possible ways this could be incorporated into deep time work, e.g. through more sophisticated emulation based on GCMs run with and without DGVMs. We also note that this is unlikely to change our main conclusions, as this positive feedback is already somewhat accounted for in the FOAM climate model simulations – which used the LPJ-DGVM – so give a 'maximal' inland hydrology throughout the SCION timeframe.

Details:

L. 129. Missing .

Added.

L. 135 There is little difference in the Cenozoic Neogene results.

That is correct. Noted.

L. 141. How is “Limited water circulation during the time of Pangaea“ calculated? Or is this an implicit assumption?

We have further clarified that SCION includes a GCM climate emulator based on the FOAM climate model.

L165. Jargon/ill-defined term. What is “biogeochemistry” referring to here: “However, the global effect of plants on biogeochemistry [...]”

We have now specified ‘chemical weathering’ here.

Reviewer #1 (Remarks to the Author):

The authors convincingly answered my doubts and questions. I think the manuscript has made a significant improvement compared to the first version, it is a topic that continues to have relative originality, therefore, I recommend its publication. Luiz Felipe Rezende.

Reviewer #2 (Remarks to the Author):

The Authors have done a very good job of adjusting the manuscript in response to my review. I recommend this article for publication.

Reviewer #3 (Remarks to the Author):

1.

The central observation, corroborated by Reviewer 1, revolves around the arid climatic conditions prevalent in the continental interiors of Pangea, especially during the Jurassic epoch. These conditions significantly influenced vegetation cover and, consequently, the removal of carbon dioxide (CO₂).

Earlier research by Donnadieu et al. (2004) employed the GeoClim climate model to demonstrate that "the breakup of Pangea causes a transition from a generally dry climatic regime during the Triassic to a wetter one during the Cretaceous." The notion of drier continental interiors is not novel. However, Gurung et al. posit that the absence of vegetation, rather than aridity alone, is responsible for elevated CO₂ levels. To support this claim, it is crucial to ascertain the extent to which weathering is constrained by runoff or vegetation (biotic enhancement of weathering). In other words, the current iteration of the manuscript lacks sufficient evidence to substantiate the core argument that a deficiency in vegetation led to higher Jurassic CO₂ levels than previously hypothesized.

Furthermore, several additional concerns necessitate attention.

2.

L80. How can it be asserted that the SCION FLORA model provides a satisfactory prediction of terrestrial biomass when Figure 2's model output omits the boreal forest (~4000 g/m²) in Siberia and Canada?

3.

It appears from the text that an assumption has been made regarding the absence of forests when temperatures fall below -10°C. Has this limit been specified as the annual average or as a threshold detrimental to forests if reached at any point during the year? In either case, this implies that the model lacks precise calibration in contemporary settings. Consequently, what justifies our confidence in its ability to forecast historical conditions?

4.

L90. What is meant by "Overall organic C burial"? Does the model account for a reduction in marine organic carbon (C) burial during periods of elevated terrestrial organic carbon (C) burial (resulting from lower nutrient concentration in oceanic supply from the continents)?

5.

The model predicts a concentration of 3000-4000 parts per million (ppm) of carbon dioxide (pCO₂) during the early Cretaceous period—a range three to five times higher than indicated by proxy data. Notably, there is no discussion of this apparent discordance. The question arises as to the basis for placing trust in the new model, particularly considering its superior alignment with data from specific periods compared to a preceding model, notwithstanding its lack of alignment with data during other temporal intervals. What attributes or qualities justify the credibility of the new model under such circumstances?

6.

What is the rationale behind the GCM emulator forecasting a lack of runoff in extensive coastal regions within the subtropical zones of Pangea?

Details

Acronyms should be described upon their first appearance, including LPJ-DGVM.

L77. Jargon such as "tissue and growth respiration" should be avoided.

Mention in parentheses all the SCION biogeochemical variables (T, runoff, insolation, CO₂, O₂, etc ??) and the FLORA variables (runoff, T, CO₂, ???).

Response to reviewer #3 for “Geographic range 1 of plants drives long-term climate change” by Gurung et al. Reviewers points are in black and responses are in blue.

1. The central observation, corroborated by Reviewer 1, revolves around the arid climatic conditions prevalent in the continental interiors of Pangea, especially during the Jurassic epoch. These conditions significantly influenced vegetation cover and, consequently, the removal of carbon dioxide (CO₂). Earlier research by Donnadieu et al. (2004) employed the GeoClim climate model to demonstrate that "the breakup of Pangea causes a transition from a generally dry climatic regime during the Triassic to a wetter one during the Cretaceous." The notion of drier continental interiors is not novel. However, Gurung et al. posit that the absence of vegetation, rather than aridity alone, is responsible for elevated CO₂ levels. To support this claim, it is crucial to ascertain the extent to which weathering is constrained by runoff or vegetation (biotic enhancement of weathering). In other words, the current iteration of the manuscript lacks sufficient evidence to substantiate the core argument that a deficiency in vegetation led to higher Jurassic CO₂ levels than previously hypothesized.

We agree that it is crucial to ascertain the extent to which weathering is constrained by runoff versus vegetation. As the reviewer notes, the direct role of runoff in changing global weathering rates has already been well documented in the GEOCLIM model. The SCION model, which we use here, is in part based on the GEOCLIM model and was co-developed by GEOCLIM developers Donnadieu and Godderis, and it represents the state of the art in modelling the relationship between runoff and weathering. Our key aim in this paper is to contrast the effect of runoff alone (default SCION model) with the effect of runoff plus vegetation change (SCION + FLORA vegetation model). To address the reviewer’s concern, we have now plotted the biotic weathering amplification factor explicitly in Figure 2, which shows that weathering is limited in more arid areas due to both a lack of runoff and lower biotic enhancement. We have also added further justification for the assumed relationship between NPP and weathering in the SI, and cited previous studies that aim to calculate the magnitude of biotic enhancement (Figure S5; lines 269-271).

Overall we believe our study brings novelty and builds upon existing knowledge in these ways: 1) SCION contains a state of the art representation of how runoff and other climate factors impact weathering and biogeochemistry at the global scale; 2) addition of a spatial vegetation model (FLORA) allows us to dynamically couple the vegetation, climate and carbon cycle for the first time: climate changes can drive vegetation change, and that change in vegetation can then impact the carbon cycle and climate – this has not been done before; 3) we produce and test a new hypothesis that the geographic range of vegetation is an important long-term climate forcing, which mostly resolves previous model-data mismatch in the Mesozoic, and may also be important for other periods of Earth history outside of the studied time interval.

We have more clearly explained these ideas in the revision and thank the reviewer for improving the work (lines 58-61).

Furthermore, several additional concerns necessitate attention.

2. L80. How can it be asserted that the SCION FLORA model provides a satisfactory prediction of terrestrial biomass when Figure 2's model output omits the boreal forest (~4000 g/m²) in Siberia and Canada?

This line refers to reference 15 in which the FLORA vegetation model was validated using a high-resolution present-day climatology and comparison to present day vegetation. In this test it reproduced the boreal forest and we consider the FLORA model to be validated against the present day in that published work. The SCION model used in this work uses a lower-resolution climate module (FOAM) rather than climate data and its focus is million-year timescale dynamics over deep time: its 'present day' scenario which is shown as 0 Ma in Figure 2 (and shown in Fig 4 of the SCION paper – Mills et al., 2021, Gondwana Res.) is more like the LGM than the Holocene and has very low northern hemisphere temperatures which prevent the growth of vegetation. We apologise for any confusion here and have made this clearer in the revision (lines 118-120)

3. It appears from the text that an assumption has been made regarding the absence of forests when temperatures fall below -10°C . Has this limit been specified as the annual average or as a threshold detrimental to forests if reached at any point during the year? In either case, this implies that the model lacks precise calibration in contemporary settings. Consequently, what justifies our confidence in its ability to forecast historical conditions?

This is an annual average and we have now stated this. As above, calibration of FLORA in contemporary settings was carried out in ref 15 using present day climatology and vegetation maps. We have been clearer in the revision that SCION is a geological timescale model and that the '0 Ma' snapshot that we show does not produce a specific representation of the current ~ 10 kyr interglacial, falling closer to the LGM at ~ 20 ka. This particular feature does not impact the overall CO_2 and temperature trend we observe over the Mesozoic and Paleogene as large land masses with $< -10^{\circ}\text{C}$ only start to occur from ~ 30 Ma onwards in the model. We have noted this in the revision.

4. L90. What is meant by "Overall organic C burial"? Does the model account for a reduction in marine organic carbon (C) burial during periods of elevated terrestrial organic carbon (C) burial (resulting from lower nutrient concentration in oceanic supply from the continents)?

We have clarified this to 'total marine and terrestrial organic carbon burial', and yes these dynamics are present within the SCION model and we have noted this (Lines 281-282).

5. The model predicts a concentration of 3000-4000 parts per million (ppm) of carbon dioxide (pCO_2) during the early Cretaceous period—a range three to five times higher than indicated by proxy data. Notably, there is no discussion of this apparent discordance. The question arises as to the basis for placing trust in the new model, particularly considering its superior alignment with data from specific periods compared to a preceding model, notwithstanding its lack of alignment with data during other temporal intervals. What attributes or qualities justify the credibility of the new model under such circumstances?

We apologise for any confusion here. The model mean predictions for the Cretaceous (green line in Figure 3) did not exceed 2000 ppm at any point and the model envelope overlaps the proxy data window. We discuss this difference in lines 198-204. We also note that the CO_2 proxies are plotted only as central estimates without their individual uncertainty windows (lines 146-147), so overlap is artificially narrowed in this way. Further discussion of proxies are on lines 148-154 and we also tried to quantify this difference numerically in line 160. If the reviewer is referring to the original SCION model (without spatial vegetation), then as we have suggested in the paper, the lack of a spatially-

resolved land biosphere may be why the previous model CO₂ was predicted to be overly high in the early Cretaceous. We have made this clearer in the revision.

6. What is the rationale behind the GCM emulator forecasting a lack of runoff in extensive coastal regions within the subtropical zones of Pangea?

Subtropical west-coast regions like NW and SW Africa, as well as the west coast of Australia, are arid at the present day due mostly to atmospheric circulation. They are typically areas of descending dry air and/or easterly trade winds which together limit moisture transport. Higher resolution GCM runs and lithological indicators of climate tend to show that this basic idea holds for paleocontinental configurations. We have noted this in the revision (lines 176-177) and thank the reviewer for the useful point.

Details

Acronyms should be described upon their first appearance, including LPJ-DGVM.

Added

L77. Jargon such as "tissue and growth respiration" should be avoided.

We have simplified this to just say respiration

Mention in parentheses all the SCION biogeochemical variables (T, runoff, insolation, CO₂, O₂, etc ??) and the FLORA variables (runoff, T, CO₂, ???).

We now state these variables, lines 247-248